# Hypoglycemia prevention practice and associated factors among diabetes mellitus patients in Ethiopia: Systematic review and meta-analyssis

**Tadele Lankrew Ayalew** [1]*, **Belete Gelaw Wale**[2], **Bitew Tefera Zewudie**[3]

**1** Department of Nursing, School of Nursing, College of Health Science and Medicine Wolaita Sodo University, Sodo, Ethiopia, **2** Department of Pediatrics and Child Health Nursing, School of Nursing, College of Health Science and Medicine Wolaita Sodo University, Sodo, Ethiopia, **3** Department of Nursing, College of Health Science and Medicine, Wolkite University, Wolkite, Ethiopia

* tadelelankrew@gmail.com

**Data Availability Statement:** All relevant data are within the paper and Supporting Information files.

**Funding:** The authors received no specific funding for this work.

## Abstract

### Background

Hypoglycemia is an urgent, life-threatening condition that requires prompt recognition and treatment for diabetes mellitus patients to prevent organ and brain damage. Hypoglycemia is one of the most important complications of diabetes mellitus patients around the globe. Hypoglycemia may increase vascular events and even death, in addition to other possible detrimental effects. In spite of the absence of other risk factors, patients receiving intensive insulin therapy are more likely to experience hypoglycemia. To reduce the risk of hypoglycemia and calculate the combined prevalence of hypoglycemia prevention practices among diabetes mellitus patients, recognition of hypoglycemia is critical.

### Objective

The main aim of this review was to evaluate the available data on Ethiopian diabetes mellitus patients' practices for preventing hypoglycemia and related factors.

### Methods and materials

This review was searched using PubMed, the Cochrane Library, Google, Google Scholar, and the Web of Sciences. Microsoft Excel was used to extract the data. All statistical analyses were done using STATA Version 14 software with a random-effects model. The funnel plot and heterogeneity of the studies were checked. Subgroup analysis was done with the study area and authors' names.

### Results

In this systematic review, 12 studies totaling 3,639 participants were included. The estimated overall practice for preventing hypoglycemia among diabetic patients in Ethiopia were 48.33% (95% CI (28.21%, 68.46%, $I^2$ = 99.7%, p ≤ 0.001). According to the subgroup analysis based on region, the highest estimated prevalence of the prevention practice of

**Competing interests:** The authors have declared that no competing interests exist.

**Abbreviations:** CDC, Centers for Disease Control and Prevention; WHO, World Health Organization; SNNPR, Southern Nation, Nationalities and Peoples' of Region.

hypoglycemia among diabetes patients in Addis Ababa was 90%, followed by SNNRP at 76.18% and in the Amhara region at 68.31% respectively. The least prevalent was observed in the Oromia region 6.10%. In this meta-analysis, diagnoses with type II diabetes (AOR = 2.53, 95%CI: 1.05, 4.04), duration (AOR = 5.49, 95%CI:3.27,7.70), taking insulin for a long time(AOR = 4.31,95%CI:2.60,6.02), having good prevention knowledge (AOR = 2.89, 95% CI: 1.15,4.23), and occupation (AOR = 4.17, 95%CI: 2.20, 6.15) were significantly associated with hypoglycemia prevention practice.

## Conclusions

This systematic review revealed that diabetic patients in Ethiopia had poor hypoglycemia prevention practices. Being an employee, taking insulin for a long time, having a good prevention practice, and having a type of diabetes mellitus were strongly correlated with practicing hypoglycemia prevention. This review implied the subsequent need for educational interventions for an individualized patient.

## Introduction

Hypoglycemia is an emergency life-threatening condition that requires prompt recognition and treatment to prevent organ and brain damage [1]. Hypoglycemia is one of the most significant complications experienced by people with diabetes mellitus worldwide [2]. Hypoglycemia is defined as an abnormally low plasma glucose concentration (70 mg/dl); however, signs and symptoms may not appear until plasma glucose concentration drops below 55 mg/dl which exposes the patient to potential harm [1, 3]. Hypoglycemia is an acute medical situation that occurs when blood sugar falls below the recommended level [4]. Along with other potential negative consequences, hypoglycemia may increase vascular events, including death [5]. Despite the absence of additional risk factors, patients receiving intensive insulin therapy are at increased risk of hypoglycemia. Depending on how severe or long-lasting it is, hypoglycemia can result in serious morbidity or even death [6]. Hypoglycemia is thought to be the cause of death for 2–4 percent of people with diabetes mellitus. It could explain the "dead-in-bed syndrome," which refers to the mysterious deaths of type 1 diabetics that happen at night [4, 7]. The signs of low blood sugar levels can change over time and vary from person to person [8]. A person with low blood sugar may experience trembling, sweating, hunger, and anxiety in the early stages. Walking difficulties, weakness, visual disturbances, strange behavior, personality changes, confusion, and unconsciousness or seizures may all be seen as the symptoms worsen [9].

The high cost of hospitalization, frequent use of ambulances, and frequent emergency room visits caused by hypoglycemia place a significant financial strain on the healthcare system. Despite being widely acknowledged by patients and their healthcare professionals, hypoglycemia is now understood to be a significant and potentially curable cause of morbidity, mortality, high costs, decreased productivity, and poor quality of life [10]. The biggest factor that makes it challenging for diabetic patients to maintain their blood sugar levels within a normal range is hypoglycemia. It had a detrimental effect on work productivity, health care resource use, and quality of life. Dietary control, taking medications as prescribed at the right times, engaging in regular physical activity, and self-monitoring blood glucose levels are all part of the prevention of hypoglycemia. Patients with diabetes should also wear identification bracelets or tags properly [11].

For both type 1 (T1DM) and type 2 (T2DM) diabetes mellitus, hypoglycemia affects patient safety and glycemic control during insulin therapy (T2DM). Because of its effects on the heart, blood vessels, eyes, kidneys, and nerves, diabetes mellitus is a leading cause of morbidity and mortality worldwide [12]. The risk of hypoglycemia is a barrier to optimal treatment of type 1 diabetes (T1DM) and type 2 diabetes (T2DM), especially in the context of insulin therapy, making blood glucose control optimization challenging [13]. The short-and long-term complications include neurologic damage, trauma, cardiovascular events, and death [14, 15]. Identification and prevention of hypoglycemia can lower the burden of diabetes by preventing hypoglycemia complications because severe untreated hypoglycemia can result in a significant economic and personal burden [1, 6, 16]. For diabetic patients, hypoglycemia is a fact of life [17]. All patients who take insulin have had hypoglycemic episodes in about 90% of cases. Hypoglycemia has been linked to between 2–4% of deaths in this population, according to estimates. The only way to determine whether a person is experiencing low blood glucose is to check blood glucose levels [18, 19].

Effective methods for lowering the risk of hypoglycemia include patient education and self-monitoring of blood glucose (SMBG), dietary changes, and regular exercise, medication alterations, careful glucose monitoring by the patient, and attentive clinician follow-up [1, 20, 21]. Additionally, Ethiopian diabetes patients should practice hypoglycemia prevention because it makes people more likely to have better habits. In a different study carried out in Ethiopia, 63.2% of participants practiced effective hypoglycemia prevention techniques. They suggested patient education as a tactic for improving the practice of hypoglycemia prevention [4, 16, 22]. Despite the fact that many studies on hypoglycemia prevention methods and awareness have been published, no estimated pooled studies have been conducted on practices and related factors regarding hypoglycemia prevention among diabetes mellitus patients in Ethiopia. Therefore, the purpose of this review was to evaluate the combined prevalence of prevention practices and related variables among patients with diabetes mellitus in Ethiopia.

## Methods and materials

### Study design and search strategy

We searched studies of hypoglycemia prevention practice and associated factors among diabetic patients in Ethiopia by using Google Scholar, African Journal online and MEDLINE, Pub Med, and Cochran Library, and Ethiopian University Repository online. We check the database at (http://www.library.ucsf.edu) and the Cochrane library to ensure this had not been done before and to avoid duplication. We also checked whether there was any similar ongoing systemic review and meta-analysis in the PROSPERO database registration message on CRD [287162]; reassured that there had been no previous similar studies undertaken. Studies published from 2015 to February 2022 were included in this review. The reference lists already identified were screened to retrieve articles. Articles were searched using MESH terms on the hypoglycemia prevention practice and associated factors among diabetic patients in Ethiopia. We strictly follow Preferred Reporting Items for Systematic and Meta-Analysis (PRISMA) protocols to estimate the hypoglycemia prevention practice and associated factors among diabetic patients in Ethiopia.

### Eligibility criteria

**Inclusion criteria.**    Eligible articles for this systematic review and meta-analysis were studies that assessed the prevention practice of hypoglycemia and associated factors among diabetic patients in Ethiopia, and studies that assessed factors associated with the prevention

practice of hypoglycemia. It included participants who are living in Ethiopia, cross-sectional study design; studies published in the English language for data analysis were included.

**Exclusion criteria.** Articles were excluded if they were: studies were done outside of Ethiopia, case reports, RCT, and reviews. An attempt was made to contact the corresponding authors using the email address or phone number as provided in the published articles.

## Data extraction and quality assessment

The parameters used in the data extraction template included the author's name, sample size, study region, study design, year of publication, OR, and 95% CI. We gathered pertinent data from accepted articles using the typical Microsoft Excel spreadsheet. The authors (BG, TL, and BT) carried out data extraction from the included articles independently. Disputes were settled through discussion and agreement with the other authors. After summarizing the relevant articles in the table and performing a critical analysis using the Joanna Brings Institute Meta-Analysis of Statistics Assessment and Review Instrument (JBI-MASTER), the studies that met the eligibility requirements were included. Joanna found studies, and their titles and abstracts were used to determine whether they should be included. Before choosing for the final review, each article was reviewed for quality. Studies with a quality assessment indicator score of seven or higher were deemed low risk (**Table 1**). Discussions with other reviewers helped resolve any disagreements that arose between them during the review process.

## Outcome variable

The first outcome of interest for this systematic review and meta-analysis was the hypoglycemia prevention practice among diabetic patients in Ethiopia. Moreover, the pooled hypoglycemia prevention practice and associated factors was computed. The second outcome variable is a factor associated with the hypoglycemia prevention practice and associated factors among diabetic patients in Ethiopia and was computed by using the log and ratio. Eighteen studies were included in this review.

**Table 1. Critical appraisal results of eligible studies in the systematic review and meta-analysis on the hypoglycemia prevention practice and associated factors among diabetic patients in Ethiopia, 2022.**

| Authors | Q1 | Q2 | Q3 | Q4 | Q5 | Q6 | Q7 | Q8 | Q9 | Total |
|---|---|---|---|---|---|---|---|---|---|---|
| Esileman A. et al | Y | Y | Y | Y | Y | N | Y | Y | Y | 8 |
| Gashayeneh G., et al | Y | N | Y | Y | Y | Y | Y | Y | Y | 8 |
| Gebrewahd B., et al | Y | Y | Y | Y | Y | Y | Y | U | Y | 8 |
| Girma N., et al | Y | Y | Y | Y | Y | Y | Y | Y | Y | 9 |
| Yitagesu M., et al | Y | Y | Y | Y | Y | Y | Y | Y | Y | 9 |
| Yohannes T., et al | Y | Y | Y | Y | N | Y | Y | Y | Y | 8 |
| Alebachew F., et al | Y | Y | Y | Y | Y | Y | U | Y | Y | 8 |
| Alemu G., et al | Y | Y | Y | Y | Y | Y | Y | U | Y | 8 |
| Solomon M., et al | Y | N | Y | Y | Y | Y | Y | Y | Y | 8 |
| Tefera K., et al | Y | Y | Y | Y | Y | Y | Y | Y | Y | 9 |
| Temesgen F., et al | Y | Y | Y | Y | Y | Y | Y | Y | Y | 9 |
| Tewodros Y., et al | Y | Y | Y | Y | Y | N | Y | Y | Y | 8 |

Y = Yes, N = No, U = Unclear; JBI Critical Appraisal Checklist for Studies Reporting Prevalence Data: Q1 = was the sample frame appropriate to address the target population? Q2.Were study participants sampled appropriately. Q3. Was the sample size adequate? Q4. Were the study subjects and the setting described in detail? Q5. Was the data analysis conducted with sufficient coverage of the identified sample. Q6.Were the valid methods used for the identification of the condition. Q7.Was the condition measured in a standard, reliable way for all participants. Q8.Was there appropriate statistical analysis. Q9. Was the response rate adequate, and if not, was the low response rate managed appropriately?

## Data processing and analysis

We used standard Microsoft Excel spreadsheet data and STATA version 14 software for data extraction and analysis of extracted data, respectively. Random effect model meta-analysis was used to compute the pooled hypoglycemia prevention practice and associated factors in Ethiopia, because eighteen studies were included in the final analysis and some of the studies used different scales, the unstandardized and standard regression coefficients of the random effect analysis model were used. Publication bias was checked by funnel plot via visual assessment. Heterogeneity between studies was checked by using Cochrane Q-statistic and $I^2$ tests. Subgroup analysis was computed to compare the hypoglycemia prevention practice and associated factors among diabetic patients within regions of Ethiopia. Point prevalence was presented in forest plot format with 95%CI.

## Results

### Identification of included studies

We found 159 articles through database searching (PubMed, Google scholar, African journal online, MEDLINE, and Cochrane library, and Ethiopian University repository online). Moreover, 151 articles reminded after removal of duplication. After reading the 151 abstracts and titles, 110 articles and 29 articles were excluded after reviewing the full articles and abstracts. Finally, 12 articles that fill the inclusion criteria were used to determine the pooled hypoglycemia prevention practice and associated factors among diabetic patients in Ethiopia (**Fig 1**).

### Characteristics of searched studies

This systematic review and meta-analysis included 3,639 study participants from 12 studies that evaluated the hypoglycemia prevention practice and associated factors among diabetic patients in Ethiopia. According to the regional distribution of the articles found through the search, three from Oromia, seven from Amhara, and one each from Tigray and Addis Ababa were included (**Table 2**).

### Hypoglycemia prevention practice among diabetic patients (systemic review and meta-analysis)

The estimated pooled prevalence hypoglycemia prevention practice among diabetic patients in Ethiopia was 48.33% (95%CI: 28.21, 68.46, $I^2$ = 99.7%, P$\leq$ 0.001) (**Fig 2**).

### Subgroup analysis of hypoglycemia prevention practice among diabetic patients in Ethiopia

We have also performed subgroup analysis based on regions where the studies were carried out. The highest hypoglycemia prevention practice was observed in Addis Ababa with a prevalence of 90.00 (95% CI: 88.54, 91.50) followed by SNNRP 76.18(72.02, 80.34) and in the Amhara region 68.31(61.58, 75.05) respectively. The least prevalent was observed in the Oromia region 6.10% (95% CI: 3.09%, 9.05%), $I^2$ = 98.70%) (**Fig 3**).

### Heterogeneity and publication bias

The $I^2$ (variation in ES attributable to heterogeneity) test results revealed that there was considerable heterogeneity with $I^2$ = 99.7%, at p-value $\leq$ 0.001. The funnel plot results revealed that a systematic distribution of the included studies through inspection, which implied there was no potential publication bias and (Egger's test: b = 0.187, p = 0.442) (**Fig 4**).

## Sensitive analysis of hypoglycemia prevention practice among diabetic patients in Ethiopia

We performed the test using a random effect, and the results showed that no single study influenced the overall pooled prevalence of hypoglycemia prevention practice in Ethiopia (**Fig 5**).

**PRISMA 2020 flow diagram for new systematic reviews which included searches of databases and registers only**

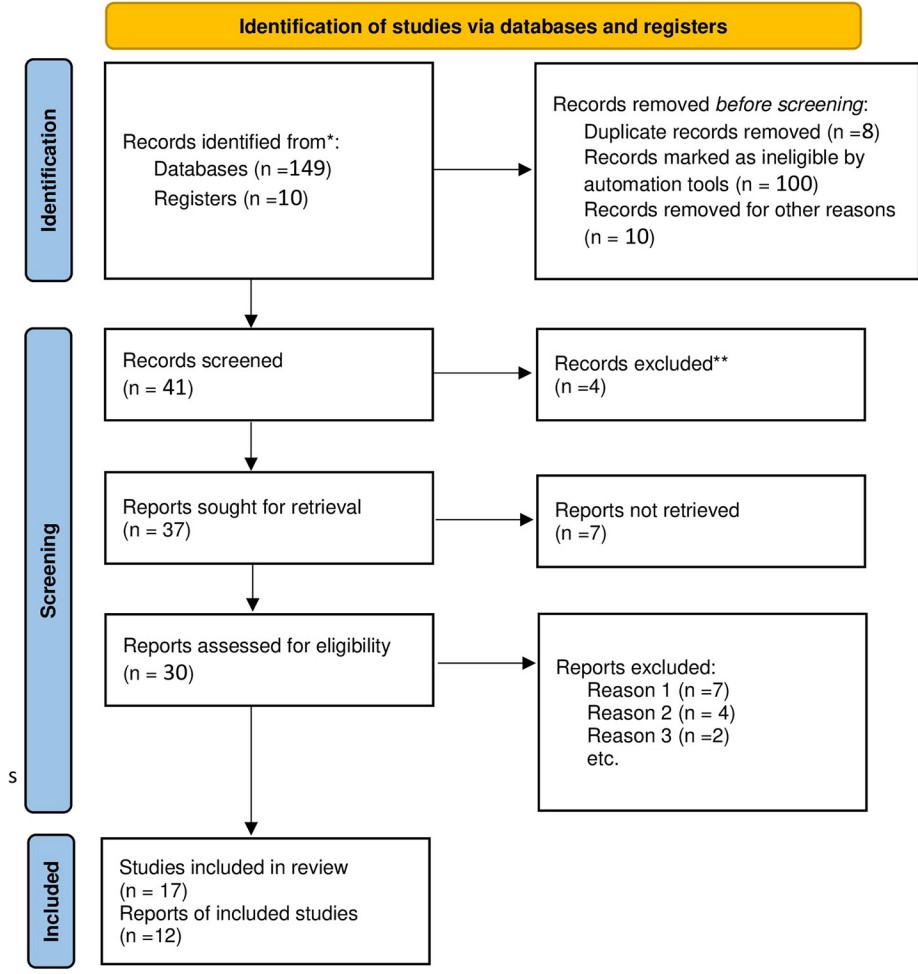

*Consider, if feasible to do so, reporting the number of records identified from each database or register searched (rather than the total number across all databases/registers).

**If automation tools were used, indicate how many records were excluded by a human and how many were excluded by automation tools.

*From:* Page MJ, McKenzie JE, Bossuyt PM, Boutron I, Hoffmann TC, Mulrow CD, et al. The PRISMA 2020 statement: an updated guideline for reporting systematic reviews. BMJ 2021; 372:n71. doi: 10.1136/bmj.n71

For more information, visit: http://www.prisma-statement.org/

**Fig 1. PRISMA diagram of selecting and including studies for systematic review and meta-analysis for the prevention practice of hypoglycaemia and associated factors among diabetic patients in Ethiopia, 2022.**

**Table 2. Characteristics of studies in a systematic review and meta-analysis on the prevalence of the hypoglycemia prevention practice and associated factors among diabetic patients in Ethiopia, 2022.**

| Authors' name | Year | Region | study area | Study period | SD | SS | cases | PP (%) |
|---|---|---|---|---|---|---|---|---|
| Esileman A. et al | 2020 | Amhara | NORTH | February-March 2019 | CS | 422 | 393 | 93.1 |
| Gashayeneh G., et al | 2019 | Amhara | East Gojjam | Nov 2017 Jan 2018 | CS | 369 | 346 | 93.7 |
| Gebrewahd B., et al | 2020 | Tigray | Public Hospitals | Mar-Apr 2018 | CS | 158 | 100 | 63.2 |
| Girma N., et al | 2015 | Amhara | South Gonder | Jun-Oct 2012 | CS | 416 | 89 | 21.4 |
| Yitagesu M., et al | 2019 | Oromia | JUMC | Apr-Jun 30, 2017 | CS | 410 | 47 | 22.90 |
| Yohanss T., et al | 2018 | Addis Ababa | TASH | Mar-Apr, 2015 | CS | 412 | 224 | 54.4 |
| Alebachew F., et al | 2019 | Amhara | UOGRTH | Feb-Mar 2017 | CS | 147 | 58 | 39.5 |
| Alemu G., et al | 2020 | Amhara | DMRH | Feb-June 2020. | CS | 423 | 311 | 73.5 |
| Solomon M., et al | 2015 | Amet al | NORTH | Jan and Feb 2013 | CS | 253 | 89 | 35.3 |
| Tefera K., et al | 2016 | Oromia | JUTH | Feb14-Apr 9, 2014 | CS | 309 | 90 | 29.1 |
| Temesgen F., et al | 2018 | Amhara | DRH | Jan1-Apr 30, 2017 | CS | 384 | 112 | 29.2 |
| Tewodros Y., et al | 2021 | Oromia | AMCH | Mar1-30, 2020 | CS | 245 | 88 | 35.9 |

NB = Hypoglycemia demarcation is <70 mg/dl; PP = prevalence percentage; SD = study design. SS = sample size

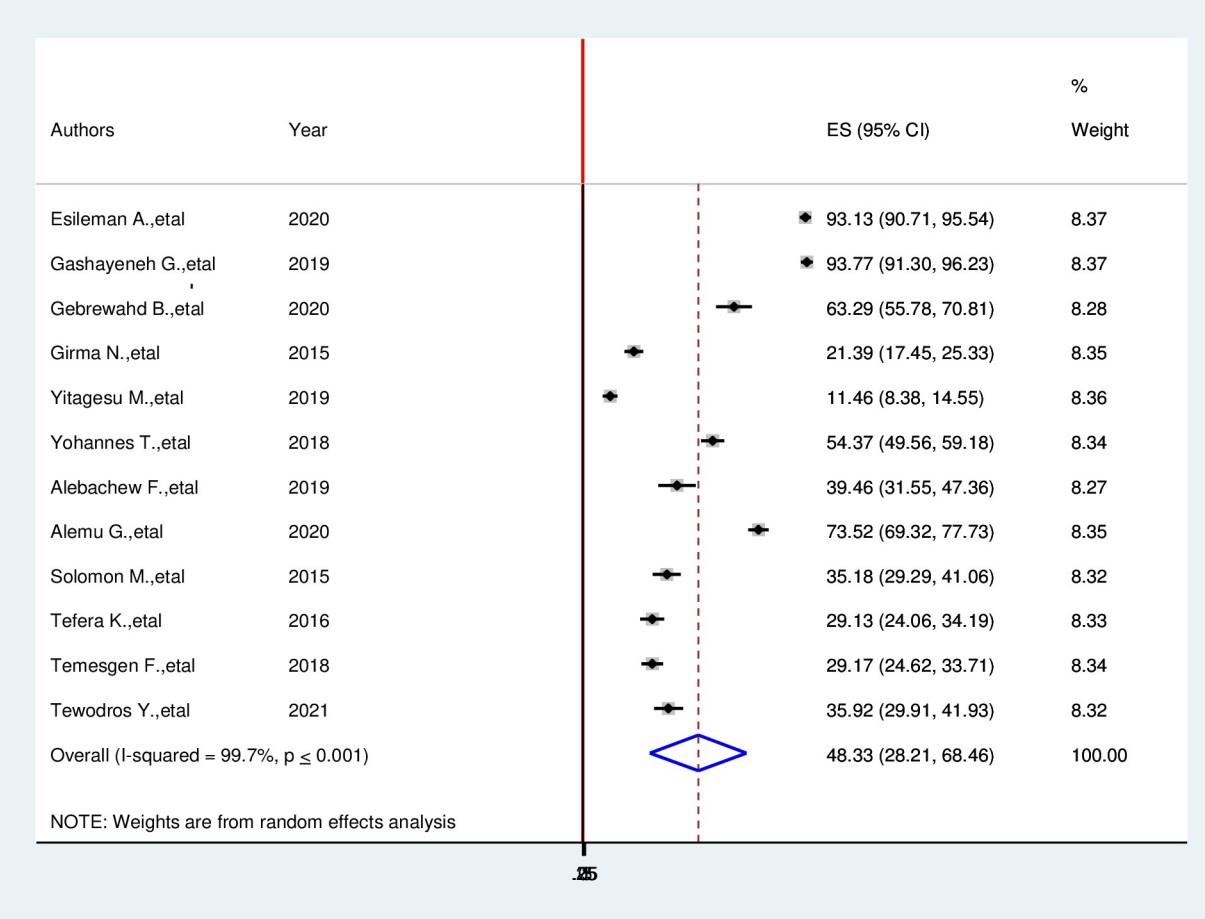

**Fig 2. Forest plot of systematic review and meta-analysis on the prevalence of hypoglycemia prevention practice and associated factors among diabetic patients in Ethiopia, 2022.**

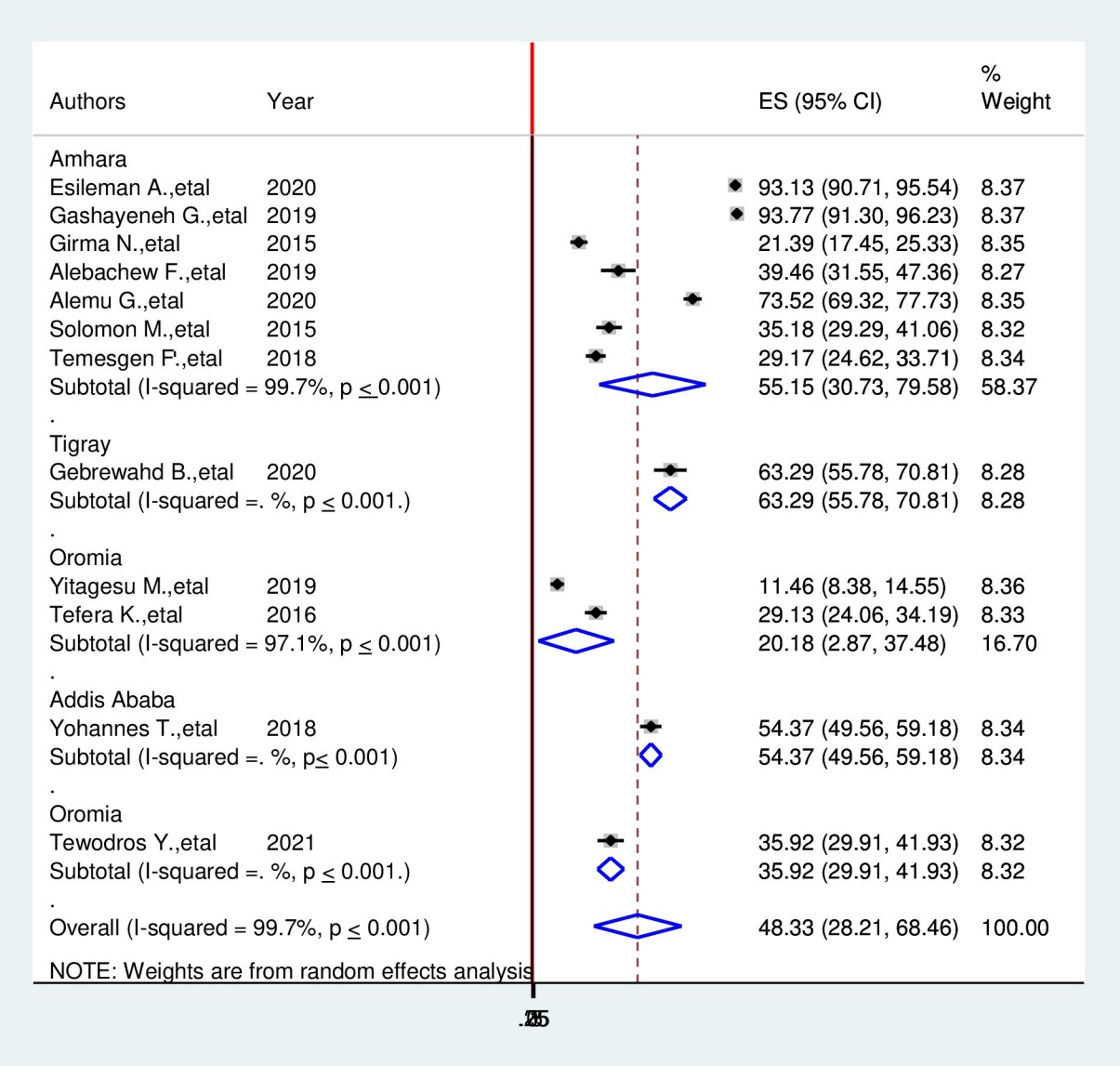

**Fig 3. Subgroup analysis of systematic review and meta-analysis on the prevalence of the prevention practice of hypoglycemia and associated factors among diabetic patients in Ethiopia, 2021.**

## Factors associated with hypoglycemia prevention practice among diabetic patients

We have comprehensively reviewed and meta-analyzed the associated factors of the hypoglycemia prevention practice and associated factors among diabetic patients in Ethiopia by using twelve relevant studies [1, 4, 16, 17, 23–30]. This study done in Ethiopia revealed the associated factors including the type of diabetes mellitus, duration of diabetes mellitus, taking of insulin therapy for a long time, good knowledge of hypoglycemia prevention practices, and occupation of the study participants in the prevention practice of hypoglycemia and associated factors among diabetic patients in Ethiopia. In this meta-analysis, diagnoses with type II diabetes mellitus were 2.53 times more likely to hypoglycemia prevention practice than persons who are

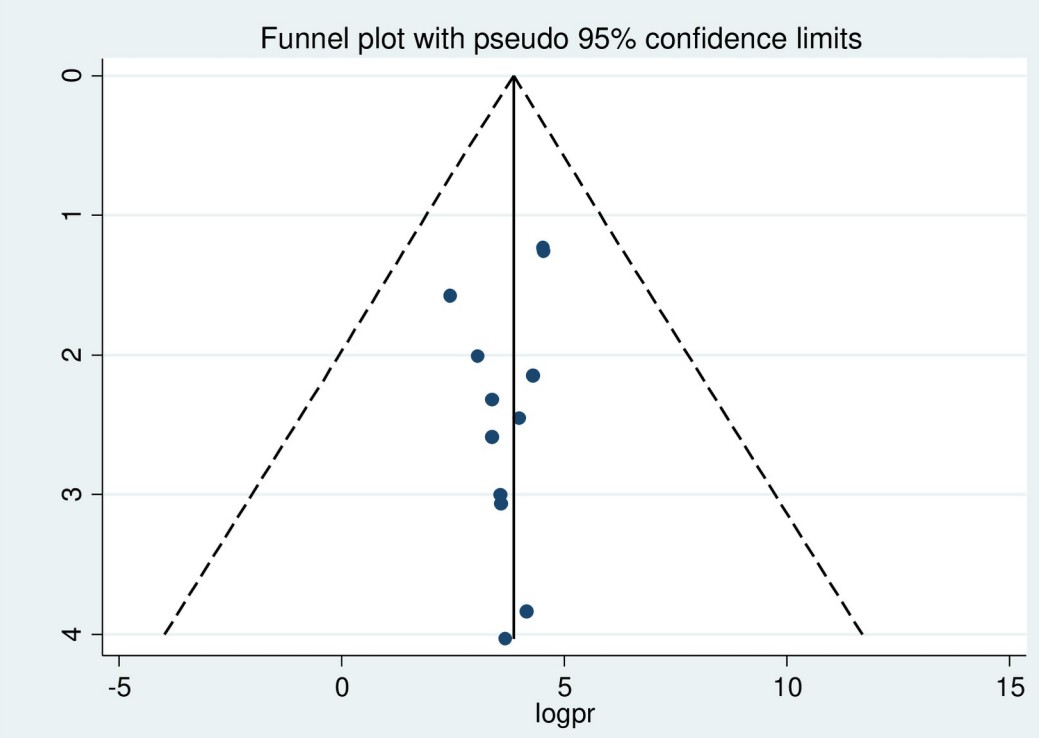

a.

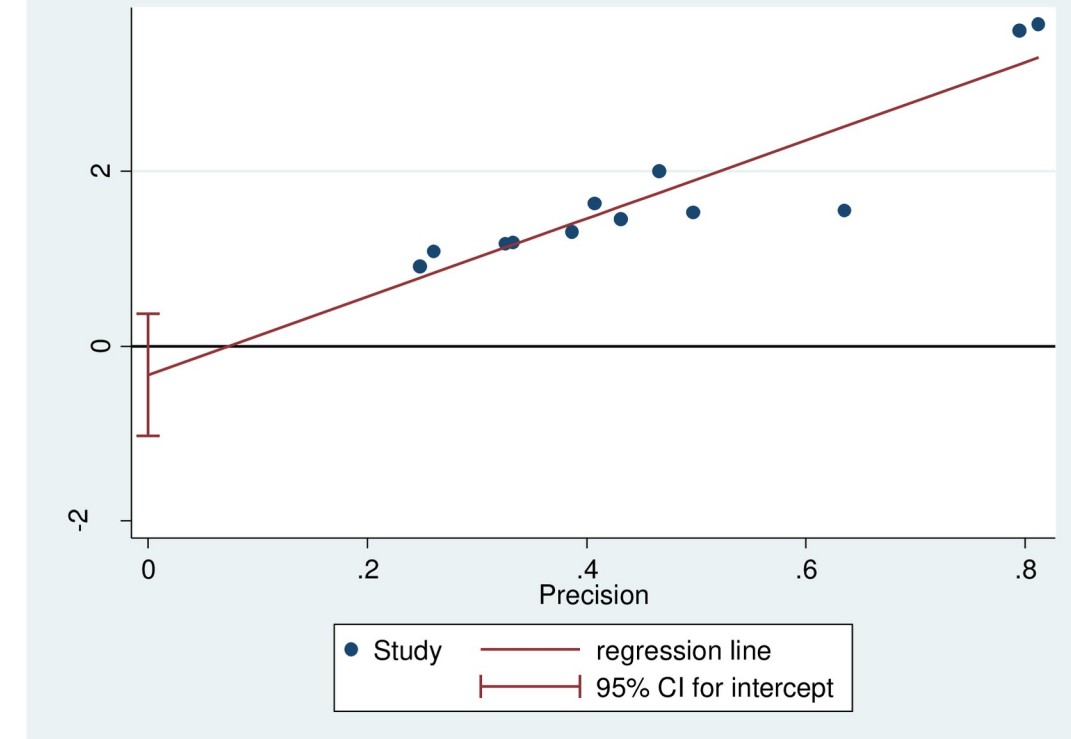

b.

**Fig 4. Funnel plot (a), Egger test (b) of a systematic review and meta-analysis on the prevention practice of hypoglycemia and associated factors among diabetic patients in Ethiopia, 2021.**

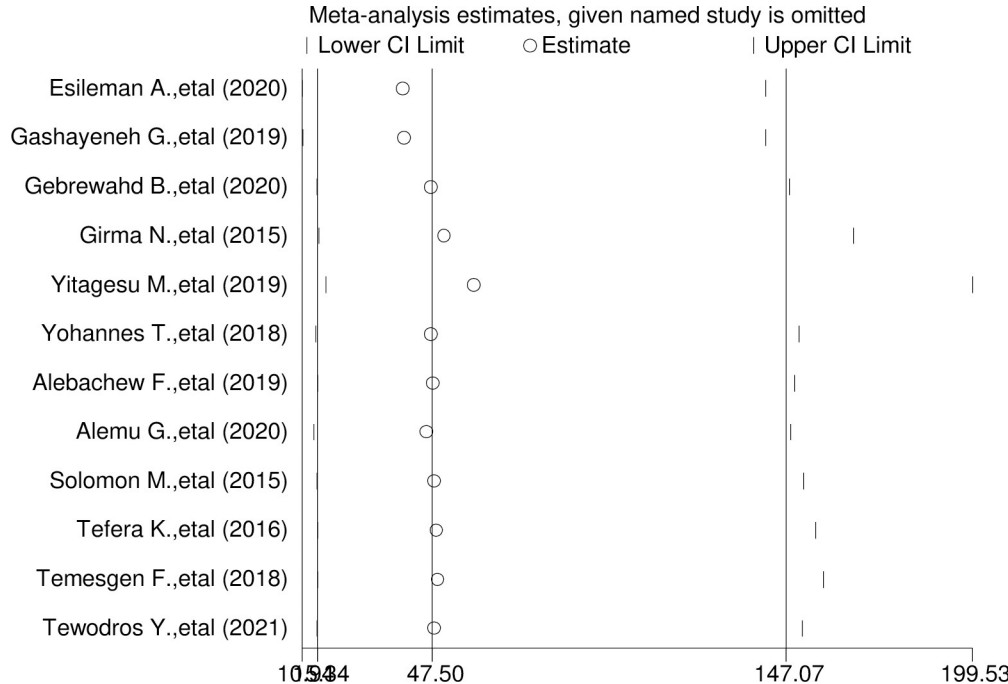

**Fig 5. Sensitive analysis of a systematic review and meta-analysis on the prevention practice of hypoglycemia and associated factors among diabetic patients in Ethiopia, 2021.**

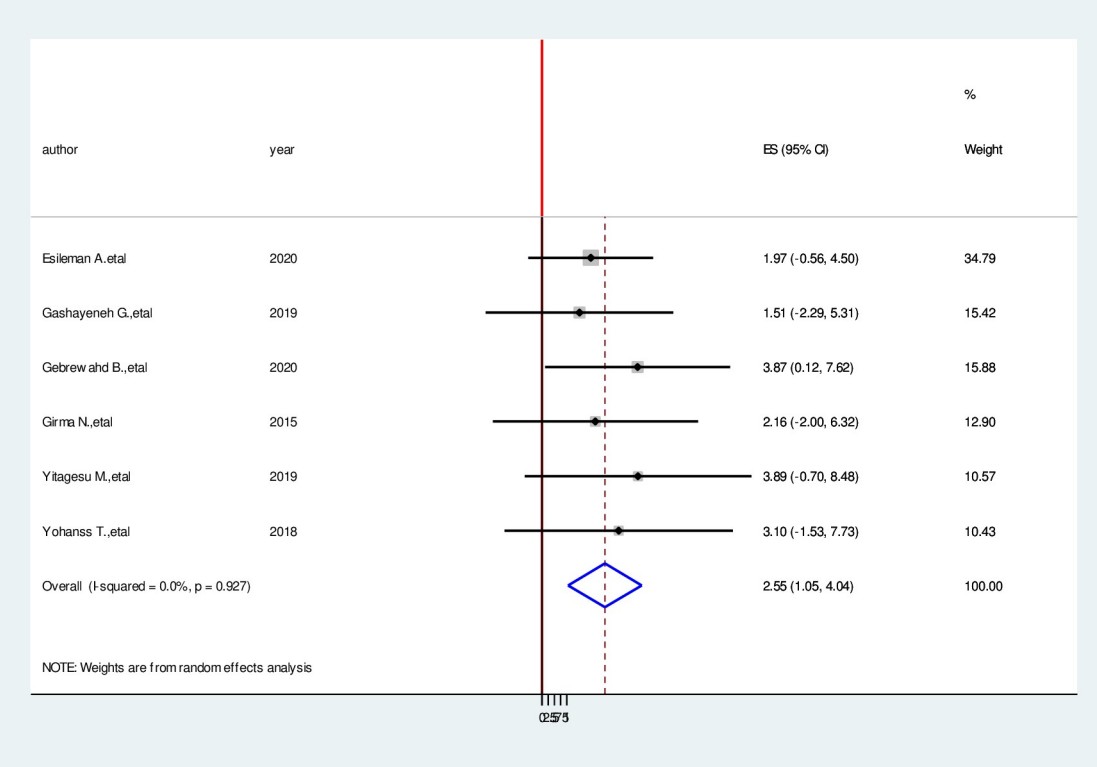

**Fig 6. Forest plot on the association between type of diabetes mellitus and hypoglycemia prevention practice among diabetic patients in Ethiopia, 2022.**

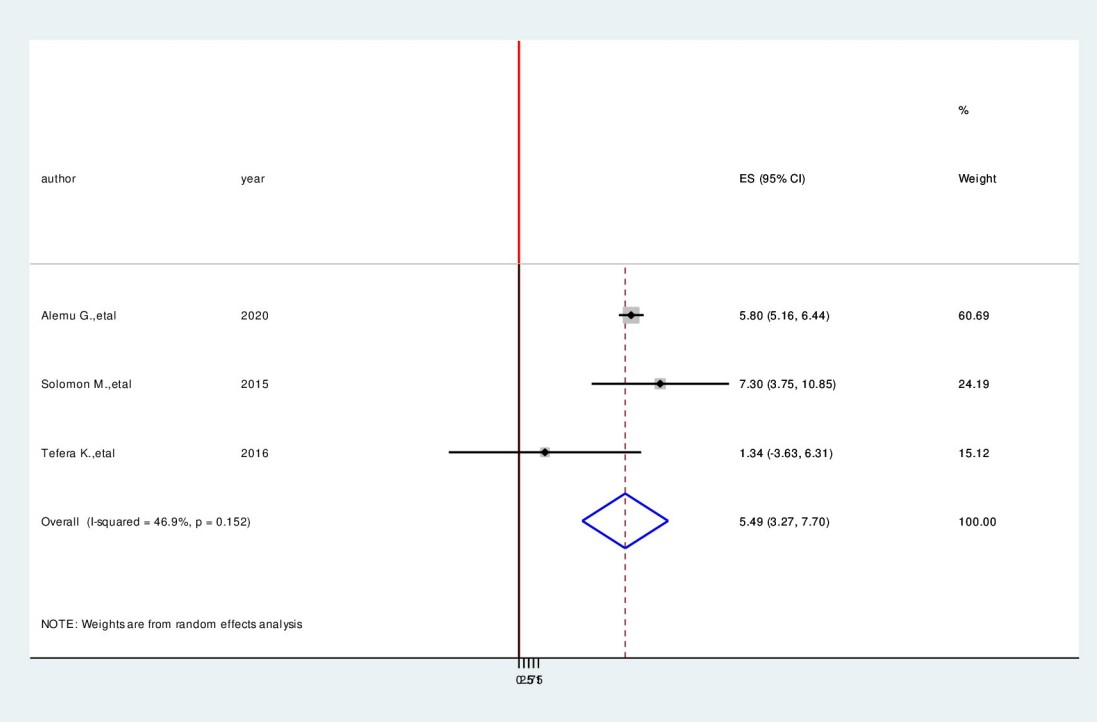

**Fig 7. Forest plot on the association between duration of diabetes milletus and hypoglycemia prevention practice among diabetic patients in Ethiopia, 2022.**

diagnosed with type I diabetes mellitus (AOR = 2.53, 95%CI: 1.05, 4.04) (Fig 6). Study participants who have more than four-year duration have 5.49 times more likely to hypoglycemia prevention practice than a person who has less than four-year duration (AOR = 5.49, 95% CI:3.27,7.70) (Fig 7). Taking insulin therapy for a longer time is 4.31 times more likely to practice prevention of hypoglycemia more than taking insulin with less time (AOR = 4.31,95% CI:2.60,6.02) (Fig 8). In this review, patients who have good prevention knowledge were 2.89 times more likely to practice prevention of hypoglycemia among diabetes mellitus patients in Ethiopia than those who have not have good knowledge (AOR = 2.89, 95%CI: 1.15,4.23) (Fig 9). The other factor significantly associated with the prevention practice of hypoglycemia is being government employment is 4.17 times more likely to practice hypoglycemia prevention (AOR = 4.17, 95%CI: 2.20, 6.15) (Fig 10).

## Discussion

In this review, the overall pooled prevalence of hypoglycemia prevention practice among diabetic patients in Ethiopia was 48.33%. This finding is lower when compared to the Ethiopian study's, which had a 49.79% accuracy rate [31], in teaching hospitals of Ethiopia 93.1% [1], in Tigray 63.2% [16]. This variation could be the result of various study factors, including different study participants, a different participant sample size, various tools, and various study designs. The output of this review is higher than the study conducted South Gondar 21.4% [4], USA 19% [32], China 31.2% [33], in the United Kingdom 45% [34]. Study period, study area, sample size differences, sampling techniques, and information sources could all serve as possible defenses.

In this meta-analysis, diagnoses with type II diabetes mellitus were significantly associated with hypoglycemia prevention practice than persons who are diagnosed with type I diabetes

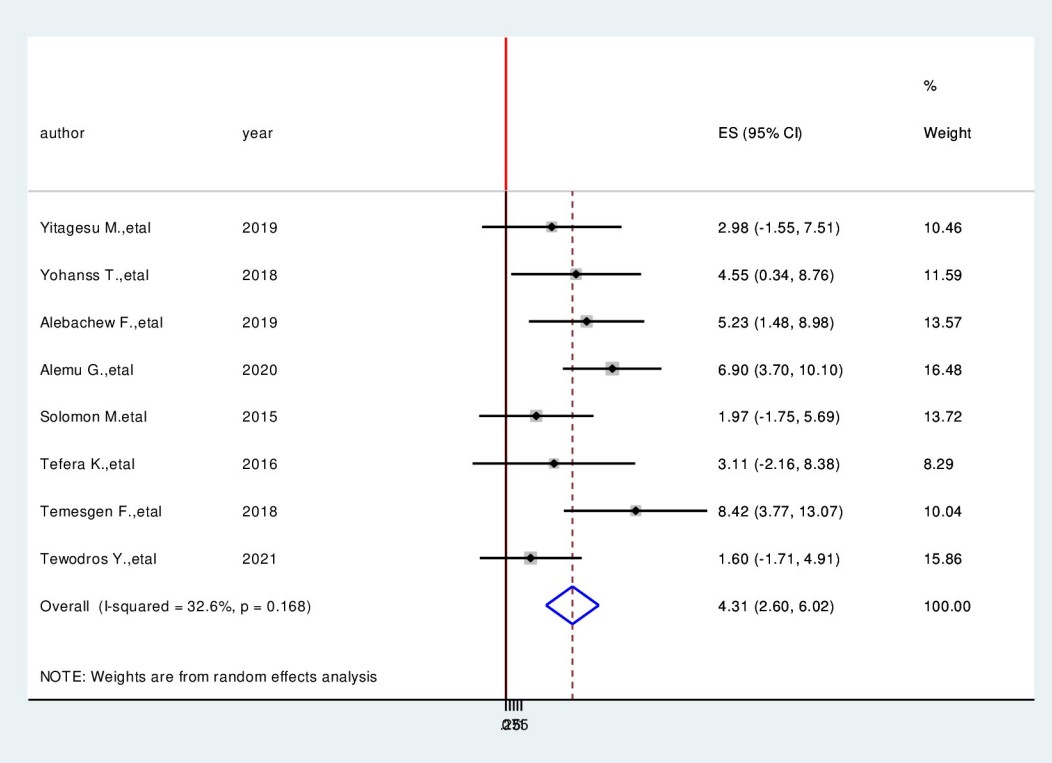

**Fig 8. Forest plot on the association between insulin taking for long time and hypoglycemia prevention practice among diabetic patients in Ethiopia, 2022.**

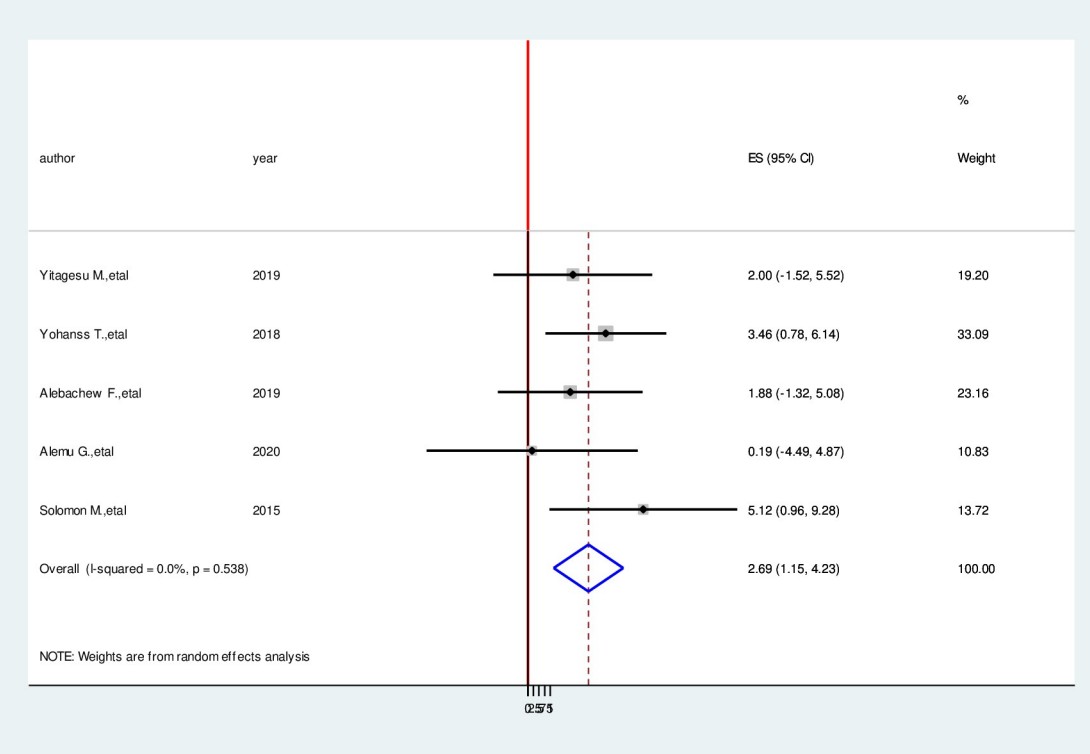

**Fig 9. Forest plot on the association between good prevention knowledge and hypoglycemia prevention practice among diabetic patients in Ethiopia, 2022.**

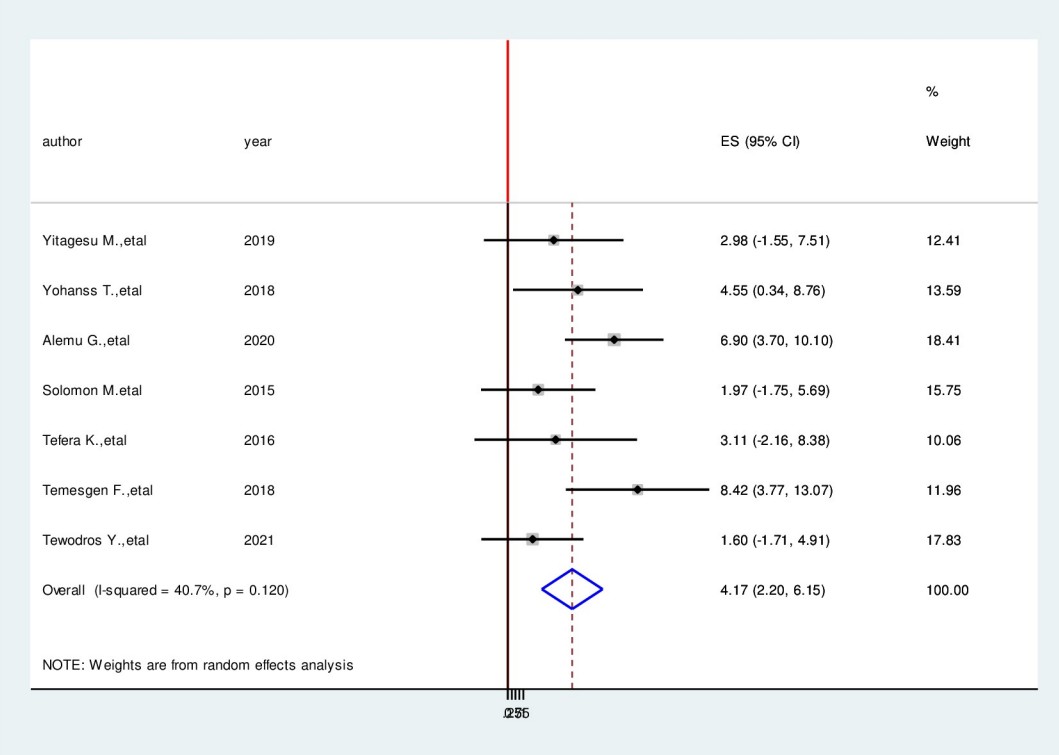

**Fig 10. Forest plot on the association between occupation and hypoglycemia prevention practice among diabetic patients in Ethiopia, 2022.**

mellitus. This study was supported by previous studies [12, 16, 33]. Study participants who have more than four-year duration have significantly associated with hypoglycemia prevention practice than a person who has less than four-year duration. This review is similar with study conducted previously [1, 16, 35]. Taking insulin therapy for a longer time is 4.31 times more likely to practice prevention of hypoglycemia more than taking insulin with less time. In this review, patients who have good prevention knowledge were 2.89 times more likely to practice prevention of hypoglycemia among diabetes mellitus patients in Ethiopia than those who have not have good knowledge. The other factor significantly associated with the prevention practice of hypoglycemia is being government employment is 4.17 times more likely to practice hypoglycemia prevention. This finding was supported by previous studies [4, 16, 35].

## Strengths of this review

The strength of this review was it uses multiple databases to search articles (both manual and electronic search) for meta-analysis, abstraction of information uniformly using a predetermined format that helped to minimize error. This meta-analysis also included studies from different parts of the Ethiopian regions.

## Limitations of this review

Bias may be there because the search was only in the English language. Primary studies of this review did not include all ten regions of Ethiopia, which may be difficult to conclude the findings for all regions in Ethiopia.

## Conclusions

In conclusion, hypoglycemia prevention practice among diabetic patients were low as revealed by this review. This review found that factors such as the type of diabetes, the length of diabetes, the use of insulin therapy for a long time, knowledge of the prevention of hypoglycemia practices, and the study participants' occupation were all significantly related to the prevention of hypoglycemia among diabetic patients in Ethiopia. This review demonstrates that hypoglycemia is surprisingly common in diabetics. Education campaigns should be incorporated on successfully spreading awareness of hypoglycemia and the best ways to prevent it in people with diabetes mellitus. In order to lower the risk of hypoglycemia, this emphasizes the need for educational interventions and the personalization of therapies.

## Supporting information

**S1 Dataset.**
(CSV)

**S1 Fig. PRISMA diagram of selecting and including studies for systematic review and meta-analysis for the prevention practice of hypoglycaemia and associated factors among diabetic patients in Ethiopia, 2021.**
(DOCX)

## Author Contributions

**Conceptualization:** Tadele Lankrew Ayalew, Belete Gelaw Wale, Bitew Tefera Zewudie.

**Data curation:** Tadele Lankrew Ayalew, Belete Gelaw Wale.

**Formal analysis:** Tadele Lankrew Ayalew, Belete Gelaw Wale, Bitew Tefera Zewudie.

**Funding acquisition:** Tadele Lankrew Ayalew, Belete Gelaw Wale.

**Investigation:** Tadele Lankrew Ayalew, Belete Gelaw Wale.

**Methodology:** Tadele Lankrew Ayalew, Belete Gelaw Wale, Bitew Tefera Zewudie.

**Project administration:** Tadele Lankrew Ayalew, Belete Gelaw Wale.

**Resources:** Tadele Lankrew Ayalew, Belete Gelaw Wale.

**Software:** Tadele Lankrew Ayalew, Belete Gelaw Wale, Bitew Tefera Zewudie.

**Supervision:** Tadele Lankrew Ayalew.

**Validation:** Tadele Lankrew Ayalew, Bitew Tefera Zewudie.

**Visualization:** Tadele Lankrew Ayalew, Bitew Tefera Zewudie.

**Writing – original draft:** Tadele Lankrew Ayalew, Bitew Tefera Zewudie.

**Writing – review & editing:** Tadele Lankrew Ayalew, Bitew Tefera Zewudie.

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
