## [Decision Letter · Decision Letter 0]

2 Aug 2022

PONE-D-22-00290Prevention practice of Hypoglycemia and Associated Factors among Diabetes Mellitus Patients in Ethiopia: Systematic Review and Meta-AnalysisPLOS ONE

Dear Dr. Lankrek

Thank you for submitting your manuscript to PLOS ONE. After careful consideration, we feel that it has merit but does not fully meet PLOS ONE’s publication criteria as it currently stands. Therefore, we invite you to submit a revised version of the manuscript that addresses the points raised during the review process.

ACADEMIC EDITOR: 

Dear dr. Lankrew

Based on several issue pointed out by the reviewers in the methodological section, I recommend a Major revision of your manuscript adressing the recomendations of the reviewers.

We look forward to receiving your revised manuscript.

Kind regards,

José Luiz Fernandes Vieira

Academic Editor

PLOS ONE

Journal Requirements:

“no”

“NO”

5. PLOS requires an ORCID iD for the corresponding author in Editorial Manager on papers submitted after December 6th, 2016. Please ensure that you have an ORCID iD and that it is validated in Editorial Manager. To do this, go to ‘Update my Information’ (in the upper left-hand corner of the main menu), and click on the Fetch/Validate link next to the ORCID field. This will take you to the ORCID site and allow you to create a new iD or authenticate a pre-existing iD in Editorial Manager. Please see the following video for instructions on linking an ORCID iD to your Editorial Manager account: https://www.youtube.com/watch?v=_xcclfuvtxQ.

6. Please include a separate caption for each figure in your manuscript.

7. Please include a copy of Table 6 which you refer to in your text on page 8.

8. Thank you for submitting the above manuscript to PLOS ONE. During our internal evaluation of the manuscript, we found significant text overlap between your submission and the following previously published works, some of which you are an author.

- https://link.springer.com/article/10.1186/2251-6581-11-17?code=676191d5-f665-426d-a036-c65ea7399639&error=cookies_not_supported

- https://journals.plos.org/plosone/article?id=10.1371%2Fjournal.pone.0238094

- https://oatext.com/pdf/IOD-1-113.pdf

- https://dmsjournal.biomedcentral.com/articles?page=5&searchType=journalSearch&sort=PubDate&tab=keyword

- https://publichealthreviews.biomedcentral.com/articles/10.1186/s40985-015-0014-z

Please revise the manuscript to rephrase the duplicated text, cite your sources, and provide details as to how the current manuscript advances on previous work. Please note that further consideration is dependent on the submission of a manuscript that addresses these concerns about the overlap in text with published work.

We will carefully review your manuscript upon resubmission, so please ensure that your revision is thorough."

Additional Editor Comments (if provided):

Dear Dr. Lankrew

Based in several issues highlighted by reviewers related to methodological issues. I reccomend a Major revision of your manuscript

Reviewers' comments:

Reviewer's Responses to Questions

**Comments to the Author**

1. Is the manuscript technically sound, and do the data support the conclusions?

Reviewer #1: Yes

Reviewer #2: Yes

2. Has the statistical analysis been performed appropriately and rigorously? 

Reviewer #1: Yes

Reviewer #2: Yes

3. Have the authors made all data underlying the findings in their manuscript fully available?

Reviewer #1: Yes

Reviewer #2: Yes

4. Is the manuscript presented in an intelligible fashion and written in standard English?

Reviewer #1: Yes

Reviewer #2: Yes

5. Review Comments to the Author

Reviewer #1: General comments

This manuscript addresses an interesting and relevant topic, and it is very important to work for Ethiopia. However, this manuscript needs substantial correction

Title

Could be Hypoglycemia prevention practice and Associated Factors among Diabetes Mellitus Patients in Ethiopia: Systematic Review and Meta-Analysis

Abstracts

In the result section of the abstract please include the meta-regression analysis result of the pooled OR with its respective CI

The sentence is not clear needs revision“ by 93.13% and 73.52% in the Amhara region respectively.”

Methods

The presentation of the search key terms is not reproducible and clearly indicated. Authors need to indicate whether title-specific or topic-specific (title-abstract-keyword) search was done in the database as well as record the entire filter applied.

Authors are advised to present the results from each database individually on the PRISMA diagram.

The choice of the time frame need to be justify as well. It is advised that the study be update to at least February 2022.

The outcome (primary and secondary) you review is not clearly stated. So, add as one sub-section in the method section of your manuscript including how you measure the primary outcome

Results

Define cases and Prevalence in your table 2. What cares and prevalence of what?

Perform a meta-regression analysis for risk factors and present your results accordingly with appropriate data presentation methods.

Discussion

The discussion is inadequate as well and too superficial.

Strength and limitation

Two independent reviewers were used in data extraction and the assessment of the risk of bias". This is a common methodological requirement for systematic reviews (PRISMA), I do not foresee this as a particular strength following a usual requirement. Authors should remove this.

Conclusion

Your conclusion is not different from your result. The content and placement of your conclusion should make its function clear without the need for additional indication of the pooled prevalence.

Missing of some abbreviations

Language editing is required. Grammatical errors and typographical errors also need to correct throughout the manuscript. Authors can give the manuscript to colleagues (senior ones) to criticize and review before resubmission

Reviewer #2: This manuscript addresses an interesting and relevant topic,and it is very important to work for Ethiopia. We know that we have very limited data. Therefore, this manuscript helps to advance the paper in the country.

Comment to the authors

1. Grammar errors need to be corrected. The manuscript presented in an intelligible fashion and written in standard English.

2. Reduce the volume of the abstraction and introduction and write the story chronologically. Currently, it is haphazard.

3. On conclusion focus only with your findings

Questions to the authors

1. How did you assess pooled prevalence?

2. what type of study designs you include in this review?

3. which study tool was used to assess review outcome?

4. How you reduce the bias of this review?

6. PLOS authors have the option to publish the peer review history of their article (what does this mean?). If published, this will include your full peer review and any attached files.

Reviewer #1: No

Reviewer #2: No

---

## [Author Response · Author response to Decision Letter 0]

18 Aug 2022

Dear Editor of PLOS ONE

This point-by-point response letter complements the responses of authors to reviewers’ comments regarding our manuscript. We are pleased to resubmit the revised version of our paper entitled “Hypoglycemia prevention practice and Associated Factors among Diabetes Mellitus Patients in Ethiopia: Systematic Review and Meta-Analysis” Tadele Lankrew Ayalew1*, Belete Gelaw Wale2, Bitew Tefera Zewudie3 which has a submission manuscript/identification number of PONE-D-22-00290R1 given by the journal. It is well known that this manuscript has been reviewed by peer reviewers and sent back to the authors for further revision as per the Journal Requirements and resubmission. We are so eager and thankful to work with you. 

We would like to take this opportunity to thank the reviewers for their views and constructive comments. The reviewers’ comments and recommendations were important to improve the quality of our manuscript. Therefore, we have organized our response letter based on reviewers’ comments and questions. (Title, Abstract, introduction, methods, results, discussion, and conclusion). Under each section, the reviewers’ comments are given followed by the authors’ responses. The authors’ responses are also shown by the track changes in the revised version of our manuscript. The responses for each of the reviewers’ comments are addressed in the following pages using the point-by-point response format. 

Our responses are written in yellow background words (highlighted with yellow).

We look forward to hearing from you at a suitable time. 

 With regards

Tadele Lankrew Ayalew

(On the behalf of all authors) 

Comments to the Authors

The authors of this systematic review and Meta-analysis study have presented valuable data to determine Hypoglycemia prevention practice and Associated Factors among Diabetes Mellitus Patients in Ethiopia. Consequently, there are specific critical issues that the reviewers would like the authors to address for further improvement in the quality of our manuscript. 

Authors’ response: we are very delighted with the reviewer’s appreciation of our efforts; and we have just given our respective responses to each of the specific reviewers concerns as detailed below. Please find below our response to the comments. All changes made in the document are highlighted in yellow word background. 

We follow PLOS ONE's style requirements, including those for file naming. Since no funding source for this review, we stated like “The authors received no specific funding for this work.”

For competing interests, because of the absence of conflict within authors, we stated, “The authors have declared that no competing interests exist."

We've checked your submission and before we can proceed, we need you to address the following issues:

1. Your article cannot proceed until you upload a copy of the completed PRISMA checklist as Supporting Information. We note that this manuscript is a systematic review or meta-analysis; our author guidelines, therefore, require that you use PRISMA guidance to help improve reporting quality of this type of study.

Thank you very much for your constructive comment. We have made corrections based on your suggestion. Please see the supporting information section.

2. Please, include a separate legend for each figure in your manuscript.

Thank you very much for your constructive comment. We have made corrections based on your suggestion. Please see line 373

3. We note that your Data Availability statement states the following: "No"

PLOS journals require that all data presented in the study be made publicly available at or before the time of acceptance. If there are legal or ethical restrictions on the data being made publicly available, such as IRB restrictions or patient confidentiality, authors must provide a way for fellow researchers to access the data.

Before we can proceed, please clarify the following:

1. Are there legal or ethical restrictions being placed upon the data? If so, please explain them in detail (e.g., data contain potentially identifying or sensitive patient information, data are owned by a third-party organization, etc.) and who has imposed them (e.g., a Research Ethics Committee or Institutional Review Board, etc.).

2. If there are no legal or ethical restrictions, please upload the data as a Supporting Information file or to a recommended stable public repository. (https://journals.plos.org/plosone/s/recommended-repositories)

Thank you very much for your constructive comment. We have made correction based on your suggestion. Please see line 267

3. Please note that PLOS does not allow authors to be the sole contact for data inquiries. If the data is only available upon request, please provide contact information, such as an email address, for a non-author, institutional point of contact (such as an IRB or ethics committee contact) who can field data inquiries from fellow researchers. If the data contact is an individual, please provide their title and relationship to the data as well.

Thank you very much for your constructive comment. Please contact:

Tesfaye Yitna Chichiabellu (Assistant Professor of Maternity and RH Nursing, School of Nursing, Wolaita Sodo University, Ethiopia.

Head of Generic Nursing Department)

Email: yefaste2005@gmail.com/tesfaye.yitna@wsu.edu.et

Cell no: +251913864251/+251930749616

OR

Tilahun  Saol Tura (Assistant Professor of Maternity and Reproductive Health Nursing, Dean of Nursing School CHSM, Wolayta Sodo University, Ethiopia)

email:tilahunsaol@gmail.com/

tilahun.saol@wsu.edu.et

Phone Number +251913734117/+251962149708

Authors’ Responses to Reviewer's Questions

Reviewers' comments and Responses to Questions

1. Is the manuscript technically sound, and do the data support the conclusions?

Reviewer #1: Yes

Reviewer #2: Yes

Thank you for your constructive comments. Really, we appreciate and accepted all the comments you raised for us. 

2. Has the statistical analysis been performed, appropriately and rigorously?

Reviewer #1: Yes

Reviewer #2: Yes

Thank you for your valuable suggestion. 

3. Have the authors made all data underlying the findings in their manuscript fully available?

Reviewer #1: Yes

Reviewer #2: Yes

Thank you for your constructive comments. We have accepted all the comments you raised.

4. Is the manuscript presented in an intelligible fashion and written in standard English?

Reviewer #1: Yes

Reviewer #2: Yes

Thank you for your constructive comments. 

5.Review Comments to the Author

Reviewer #1: General Comments

This manuscript addresses an interesting and relevant topic, and it is very important to work for Ethiopia. However, this manuscript needs substantial correction

Title

could be Hypoglycemia prevention practice and Associated Factors among Diabetes Mellitus Patients in Ethiopia: Systematic Review and Meta-Analysis

Thank you for your constructive comments. We have accepted your comments you raised. We have made correction based on your suggestion to the revised manuscript. In the revised manuscript, we have corrected like” Hypoglycemia prevention practice and Associated Factors among Diabetes Mellitus Patients in Ethiopia: Systematic Review and Meta-Analysis” ==>Please see on the title page.

Abstracts

In the result section of the abstract please include the meta-regression analysis result of the pooled OR with its respective CI

Thank you for your constructive comments. We have accepted your comments you raised. We have made corrections based on your suggestion to the revised manuscript. ==>Please, see the abstract section.

The sentence is not clear and needs revision“ by 93.13% and 73.52% in the Amhara region respectively.”

Thank you very much for your critical comments. We would like to say sorry for the unclear expression. We have accepted the comments you raised. We have made corrections in the revised manuscript. ==>Please, see the abstract section and thorough the revised manuscript.

Methods

The presentation of the search key terms is not reproducible and clearly indicated. Authors need to indicate whether title-specific or topic-specific (title-abstract-keyword) search was done in the database as well as record the entire filter applied.

Thank you for your constructive comments. We have accepted the comments you raised. We have made corrections based on your suggestion to the revised manuscript.

Authors are advised to present the results from each database individually on the PRISMA diagram.

Thank you for your constructive comments. We have accepted the comments you raised. We have made corrections based on your suggestion to the revised manuscript.

The choice of the time frame need to be justify as well. It is advised that the study be update to at least February 2022.

Thank you for your constructive comments. We have made correction based on your suggestion to the revised manuscript. ==>Please, see line 124.

The outcome (primary and secondary) you review is not clearly stated. So, add as one sub-section in the method section of your manuscript including how you measure the primary outcome

Thank you for your constructive comments. We would like to say sorry for the unclear expression. We had avoid based on your suggestion to the revised manuscript.

Results

Define cases and Prevalence in your table 2. What cases and prevalence of what?

Thank you for your constructive comments. We would like to say sorry for the unclear expression. Here our concern was that case showed that the frequency and describe in number, whereas prevalence showed that the percentage of the case. 

Perform a meta-regression analysis for risk factors and present your results accordingly with appropriate data presentation methods.

Thank you for your constructive comments. We apologize for the missing of risk factors analysis. We have made correction based on your suggestion to the revised manuscript. ==>Please, see lines 205-221.

Discussion

The discussion is inadequate as well and too superficial.

Thank you very much for your critical review. After proofreading, the appropriate correction was made to the revised manuscript based on the given comment throughout the manuscript. ==>Please, see discussion section.

Strength and limitation

Two independent reviewers were used in data extraction and the assessment of the risk of bias". This is a common methodological requirement for systematic reviews (PRISMA), I do not foresee this as a particular strength following a usual requirement. Authors should remove this.

Thank you for your constructive comments. We apologize for the unclear expression. We removed based on your suggestion to the revised manuscript. ==>Please Strength and limitation section.

Conclusion

Your conclusion is not different from your result. The content and placement of your conclusion should make its function clear without the need for additional indication of the pooled prevalence.

Thank you very much for your critical review. After proof reading, the appropriate correction was made to the revised manuscript based on the given comment throughout the manuscript. ==>Please, see conclusion section.

Missing of some abbreviations

Language editing is required. Grammatical errors and typographical errors also need to correct throughout the manuscript. Authors can give the manuscript to colleagues (senior ones) to criticize and review before resubmission

Thank you very much for your critical review. After proof reading, the appropriate correction was made to the revised manuscript based on the given comment throughout the manuscript. Concerning grammatical errors and typographical errors, we had use online grammar checker and consulted to senior expert in English language. 

Reviewer #2: This manuscript addresses an interesting and relevant topic,and it is very important to work for Ethiopia. We know that we have very limited data. Therefore, this manuscript helps to advance the paper in the country.

Comment to the authors

1. Grammar errors need to be corrected. The manuscript presented in an intelligible fashion and written in standard English.

Thank you very much for your critical review. After proof reading, the appropriate correction was made to the revised manuscript based on the given comment throughout the manuscript. Concerning grammatical errors and typographical errors, we used an online grammar checker and consulted to senior expert in the English language. 

2. Reduce the volume of the abstraction and introduction and write the story chronologically. Currently, it is haphazard.

Thank you very much for your critical review. After proofreading, the appropriate correction was made to the revised manuscript based on the given comment throughout the manuscript. 

3. On conclusion focus only with your findings

Thank you very much for your critical review. After proof reading, the appropriate correction was made to the revised manuscript based on the given comment throughout the manuscript. 

Questions to the authors

1. How did you assess pooled prevalence?

Thank you very much. We used standard Microsoft Excel spreadsheet data and STATA version 14 software for data extraction and analysis of extracted data, respectively. Finally random effect model meta-analysis was used to compute the pooled prevalence.==>please see line 165

2. what type of study designs you include in this review?

Thank you very much for valuable questions. We had used cross-sectional study.

3. which study tool was used to assess review outcome?

Thank you very much. We used Joanna Brings Institute Meta-Analysis of Statistics Assessment and Review Instrument (JBI-MASTER) and We strictly follow Preferred Reporting Items for Systematic and Meta-Analysis (PRISMA) protocols to estimate the hypoglycemia prevention practice and associated factors among diabetic patients in Ethiopia. ==> Please, see line 145.

4. How you reduce the bias of this review?

Thank you very much for encouraging us. To reduce the bias of this review, first, we set the eligible criteria, and then the funnel plot and egger’s test were used. ==> Please, see line 210.

6.PLOS authors have the option to publish the peer review history of their article (what does this mean?). If published, this will include your full peer review and any attached files.

It means PLOS now offers accepted authors the opportunity to publish the peer review history of their manuscript alongside the final article. The peer review history package includes the complete editorial decision letter for each revision, with reviews, and your responses to reviewer comments, including attachments.

---

## [Decision Letter · Decision Letter 1]

26 Sep 2022

Hypoglycemia prevention practice and Associated Factors among Diabetes Mellitus Patients in Ethiopia: Systematic Review and Meta-Analysis

PONE-D-22-00290R1

Dear Dr. Tadele Lankrew

We’re pleased to inform you that your manuscript has been judged scientifically suitable for publication and will be formally accepted for publication once it meets all outstanding technical requirements.

Kind regards,

José Luiz Fernandes Vieira

Academic Editor

PLOS ONE

Additional Editor Comments (optional):

Reviewers' comments:

Reviewer's Responses to Questions

**Comments to the Author**

1. If the authors have adequately addressed your comments raised in a previous round of review and you feel that this manuscript is now acceptable for publication, you may indicate that here to bypass the “Comments to the Author” section, enter your conflict of interest statement in the “Confidential to Editor” section, and submit your "Accept" recommendation.

Reviewer #1: All comments have been addressed

Reviewer #2: All comments have been addressed

2. Is the manuscript technically sound, and do the data support the conclusions?

Reviewer #1: Yes

Reviewer #2: Yes

3. Has the statistical analysis been performed appropriately and rigorously? 

Reviewer #1: Yes

Reviewer #2: Yes

4. Have the authors made all data underlying the findings in their manuscript fully available?

Reviewer #1: Yes

Reviewer #2: Yes

5. Is the manuscript presented in an intelligible fashion and written in standard English?

Reviewer #1: Yes

Reviewer #2: Yes

6. Review Comments to the Author

Reviewer #1: Thank you for giving me the opportunity to review this manuscript. I have seen the revised manuscript. All my previous comments and concerns, especially on the Language and grammar of the manuscript, Abstract, Methods, Results. Discussion (limitations) are adequately addressed and corrected. Thank you the authors for your detailed corrections of my comments and concerns. I believe that the manuscript is now suitable for publication and I recommend accepting it for publication. Thanks for the clear explanation and revisions. I don't think I have any additional concerns.

Reviewer #2: The authors try to address all my comments, so it is accepted for publication from my point of view.

7. PLOS authors have the option to publish the peer review history of their article (what does this mean?). If published, this will include your full peer review and any attached files.

Reviewer #1: No

Reviewer #2: No

---

## [Editor Report · Acceptance letter]

19 Oct 2022

PONE-D-22-00290R1 

Hypoglycemia prevention practice and Associated Factors among Diabetes mellitus Patients in Ethiopia: Systematic Review and Meta-Analysis 

Dear Dr. Lankrew Ayalew:

I'm pleased to inform you that your manuscript has been deemed suitable for publication in PLOS ONE. Congratulations! Your manuscript is now with our production department. 

Kind regards, 

on behalf of

Dr. José Luiz Fernandes Vieira 

Academic Editor

PLOS ONE